# Identifying Subphenotypes for Sepsis with Acute Kidney Injury via Multimodal Graph State Space Models

Haowei Xu
National Institute of Health Data
Science, Peking University
Beijing, China
hwxu@mail.nwpu.edu.cn

Tongyue Shi
National Institute of Health Data
Science, Peking University
Beijing, China
tyshi@stu.pku.edu.cn

Wentie Liu
National Institute of Health Data
Science, Peking University
Beijing, China
wtliu@stu.pku.edu.cn

Huiying Zhao
Peking University People's Hospital
Beijing, China
zhaohuiying109@sina.com

Guilan Kong*
National Institute of Health Data
Science, Peking University
Beijing, China
Advanced Institute of Information
Technology, Peking University
Zhejiang, China
guilan.kong@hsc.pku.edu.cn

## ABSTRACT

Sepsis with acute kidney injury (SAKI) is a heterogeneous clinical syndrome and a leading cause of mortality in intensive care units (ICUs). Identifying subphenotypes of SAKI can improve treatment precision, enabling more targeted clinical interventions. Recently, the analysis of sepsis subphenotypes using electronic health records (EHRs) has gained interest among healthcare researchers. However, current methods typically rely on unimodal features, overlooking intrinsic correlations among patients and struggling with the sparse and high-dimensional nature of EHR data. In this paper, we propose **MGSSM-SAKI**, a novel **M**ultimodal **G**raph **S**elective **S**tate **S**pace **M**odel for identifying subphenotypes of **SAKI**. First, we develop a multimodal fusion module that integrates demographic information, laboratory results, vital signs, and diagnostic data. Next, we introduce an adaptive latent graph inference module that captures latent graph structures and co-optimizes them with the identification model to reveal intrinsic patient connections. Inspired by the recent success of state space models (SSMs), such as Mamba, we incorporate a graph learning model that combines graph neural networks with selective SSMs. Finally, we design a spectral modularity maximization objective function with regularization terms to achieve differentiable patient subphenotype identification. Experiments conducted on the MIMIC-IV dataset demonstrate that our model outperforms baseline models, exhibiting strong performance and interpretability.

*Corresponding author.

*KDD 2024 Workshop AIDSH, Aug 26 2024, Barcelona, Spain*

## CCS CONCEPTS

• **Applied computing** → **Health informatics**; • **Social and professional topics** → **Medical records**; • **Mathematics of computing** → *Time series analysis*; • **Computing methodologies** → *Neural networks.*

## KEYWORDS

Subphenotype, Sepsis with Acute Kidney Injury, Graph Representation Learning, State Space Models

## 1 INTRODUCTION

Sepsis is defined as organ dysfunction resulting from the host's deleterious response to infection [14]. One of the most common organs affected is the kidneys, resulting in Sepsis with Acute Kidney Injury (SAKI) that contributes to the morbidity and mortality of sepsis [3, 16, 25]. Managing SAKI is particularly challenging due to the variability in patient responses to medical treatments. Identifying distinct SAKI subphenotypes could enable more precise and targeted clinical interventions [18, 19].

The significant increase in the volume and diversity of electronic health records (EHRs) over the past few decades has enabled the application of machine learning to classify patients into different subphenotypes [31]. EHRs, which include temporal sequence data, demographics, diagnoses, medications, lab results, vital signs, and other relevant information, provide a comprehensive dataset for analysis. Current SAKI subphenotyping models cluster patients by aggregating key clinical variables, such as heart rate and respiratory rate, observed during the first day of ICU stays. However, these methods primarily face the following three challenges [12].

*Failure to integrate multimodal information from EHRs.* In real diagnostic scenarios, due to the insufficiency of single-modal data to provide adequate information required for accurate diagnosis, medical experts often analyze various modalities of patient data comprehensively to make reliable diagnoses. Existing SAKI subphenotyping frameworks typically aggregate clinical variables to compute patient similarity. However, these approaches neglect the

multimodality of the variables, which is a crucial aspect of EHR data [32].

*Lack of consideration for similarities among patients.* The relationships among patients are often insufficiently considered, despite their significance in diagnosis. Graph models, by their inherent characteristics, offer a versatile method for integrating multimodal information and uncovering relationships among patients [21]. Most graph-based approaches construct a patient relationship graph from existing multimodal features using predefined similarity measures, and then apply Graph Neural Networks (GNNs) to aggregate patient features over local neighborhoods for prediction [21, 23]. This approach not only compromises the model's integrality but also results in suboptimal performance in downstream tasks. A more effective strategy involves learning the graph adaptively, an area explored to some extent in recent GNN studies [29, 35].

*Difficulty in handling sparse and high-dimensional data.* EHRs typically contain a large amount of sparse and high-dimensional data [10]. For example, the length and information content of records can vary significantly between different patients, requiring models to handle substantial amounts of empty or sparse data during training and inference [34]. This increases computational complexity and storage requirements. In recent years, State Space Models (SSMs) have gained popularity as an efficient alternative to attention-based sequence modeling architectures, such as Transformers [5, 28]. SSMs can be conceptualized as RNNs with fixed lengths that do not grow with input length, bringing significant efficiency benefits in terms of inference speed and computation/memory complexity compared to Transformers [17]. This indicates their great potential for application in EHR data.

To address the aforementioned challenges, this paper presents **MGSSM-SAKI**, a novel **M**ultimodal **G**raph **S**elective **S**tate **S**pace **M**odel for identifying subphenotypes of **SAKI**. Our key contributions are as follows:

- **Multimodal Feature Fusion.** To achieve representation learning of multimodal data, we collected demographic information, a series of diagnostic results, and multiple variable sets (including laboratory tests and vital signs) from the EHR of patients with SAKI.
- **Adaptive Latent Graph Inference**. To capture the similarities among SAKI patients, we design an adaptive latent graph inference (LGI) module. Firstly, we use the fused multimodal features of each patient as input. Next, we propose a simple yet effective learnable adjacency matrix constructor, which is jointly optimized with the downstream model. Finally, a non-negative sparse patient graph is obtained through threshold-based sparsification.
- **Graph State Space Models**. To address sparse and high-dimensional data, we utilize SSM for representation learning on patient graphs. Inspired particularly by Transformer-related works, we have designed four steps to implement our SSM: (1) *Tokenization*; (2)*Token Ordering*; (3) *Local Encoding*; (4) *Bidirectional Selective SSM Encoder*.

## 2 METHODOLOGY

This section provides a comprehensive introduction of the various components and design philosophies of MGSSM-SAKI, with the overall framework depicted in Figure 1.

(1) **Multimodal Fusion.** Multimodal information of each SA-AKI patient is embedded into a high-dimensional vector space through specific encoders, followed by irregularity-awre feature fusion.

(2) **Latent Graph Inference.** Potential similarities between patients are adaptively inferred from given multimodal node features.

(3) **Graph State Space Models.**
- **Tokenization:** The graph is mapped into a sequence of tokens.
- **Local Encoding:** Local structures around each node are encoded using GNNs.
- **Token Ordering:** The sequence of tokens is ordered based on the context.
- **Bidirectional SSMs:** Relevant nodes or subgraphs are selected and flow into the hidden states.

(4) **Patient Subphenotyping.** Subphenotypes with distinct clinical characteristics and mortality trajectories are identified.

## 2.1 Multimodal Fusion

*Time Series Embedding.* A discretized multi-time attention (mTAND) module is utilized to re-represent time series into $\alpha$ [22]. Initially, $\mathbf{x}^{ts}$ is discretized based on $\mathbf{t}^{ts}$ into hourly intervals with a sequence of regular time points, $\boldsymbol{\alpha} = [0, 1, \cdots, \alpha - 1]$. To incorporate irregular time information, a time representation method, Time2Vec [8], transforms each value in a list of continuous time points, $\tau$, of arbitrary length $l_\tau$, into a vector of size $d_v$, resulting in a series of time embeddings $\theta(\boldsymbol{\tau}) \in \mathbb{R}^{l_\tau \times d_v}$:

$$\theta(\boldsymbol{\tau})[i] = \begin{cases} \omega_i \boldsymbol{\tau} + \phi_i & \text{if } i = 1 \\ \sin(\omega_i \boldsymbol{\tau} + \phi_i) & \text{if } 1 < i \leq d_v \end{cases}, \tag{1}$$

where $\theta(\boldsymbol{\tau})[i]$ is the $i$-th dimension of Time2Vec, and $\{\omega_i, \phi_i\}_{i=1}^{d_v}$ are learnable parameters. The sine function captures periodic patterns, while the linear term captures non-periodic behaviors, conditional on the progression of time. The mTAND module leverages $V$ different Time2Vec instances, $\{\theta_v(\cdot)\}_{v=1}^V$, to produce interpolation embeddings $\mathbf{E}_t^X$ at $\boldsymbol{\alpha}$ using a time attention mechanism.

*Demographics and Diagnosis Embedding.* For patient demographics, ages are categorized into several groups (e.g., 18-30, 30-40, 40-50, etc.). Each patient's age group and gender are processed through an embedding layer, represented by an embedding matrix $\mathbf{E}^{D^e} \in \mathbb{R}^{2 \times k}$ [33]. Similarly, the embeddings for diagnoses are obtained as $\mathbf{E}^D = \left[ e_1^d, e_2^d, \ldots, e_{|D|}^d \right] \in \mathbb{R}^{|D| \times k}$.

*Multimodal Embedding Fusion.* To fuse the embedding matrices of various EHR data $\mathbf{E}^{D^e}, \mathbf{E}^D, \mathbf{E}_t^X \in R^{* \times k}$, fully connected layers are employed to project them into the same semantic space. The results are then concatenated within this new semantic space to produce a matrix $\mathbf{E}_t \in R^{* \times k}$, which encapsulates multi-modal information

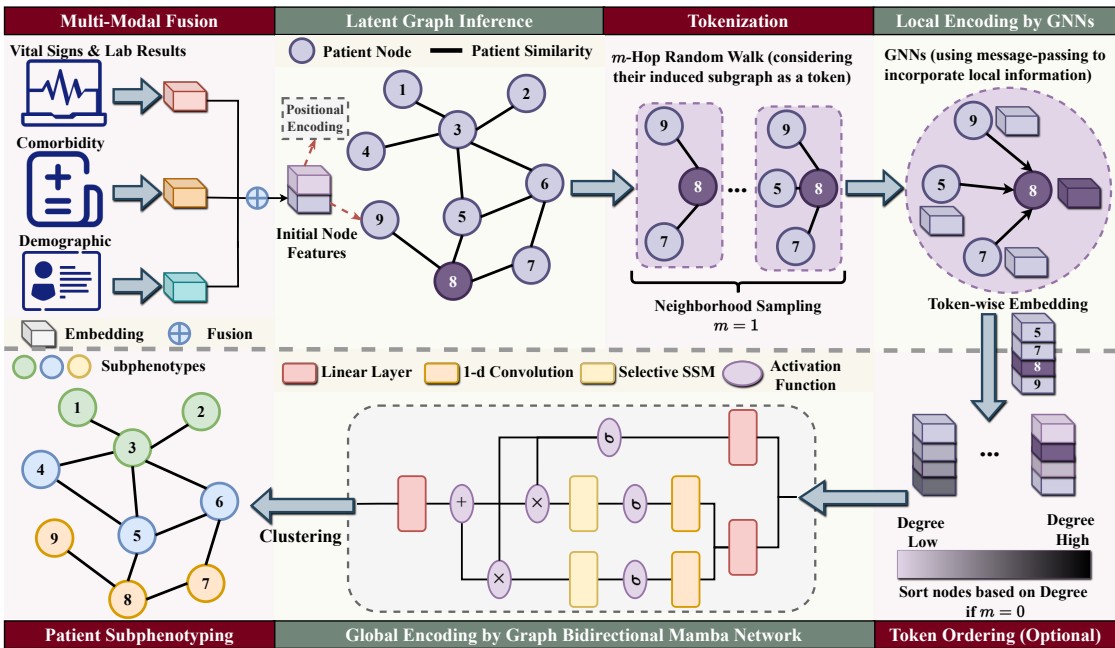

**Figure 1: Overview of our framework.**

at time $t$. Due to missing values and the varying numbers of medications at different times, the lengths of $\mathbf{E}_t$ are inconsistent. To address this, a max-pooling layer maps $\mathbf{E}_t$ to a vector $\mathbf{e}_t \in R^k$. It can be assumed that a well-trained model will ensure $\mathbf{e}_t$ retains the essential information of $\mathbf{E}_t$. Then, we input the $\mathbf{e}_t$ to a Long Short-Term Memory (LSTM) to get the final patient representation set $\mathbf{H}$.

## 2.2 Adaptive Latent Graph Inference

Graph SSMs leverage existing graph structures to learn node representations for various downstream tasks through a message-passing mechanism. In the biomedical field, however, obtaining a suitable graph for specific tasks is often challenging. Consequently, graph learning becomes a critical consideration when applying graph SSMs to biomedical tasks.

*Adjacency Matrix Learning.* Graph learning can be approached in two primary ways: (i) learning a joint discrete probability distribution on the edges of the graph [4] or (ii) learning a similarity metric for nodes [13]. Due to the non-differentiable nature and limited applicability to inductive learning of the former method, the focus here is on the latter—similarity metric learning of nodes. Previous methods have employed various similarity metrics, such as radial basis function (RBF) kernel [36], cosine similarity [9], and threshold-based metrics [6] for discrete features. However, these approaches often require meticulous manual tuning to construct a meaningful graph structure for downstream graph SSMs.

To address this issue, a simple yet effective learnable metric function is proposed, which can be optimized jointly with downstream

graph SSMs. This function is defined as:

$$\hat{A}_{ij} = \text{Sim}(h_i, h_j) = \cos(W_A^\top h_i, W_A^\top h_j), \quad (2)$$

where $W_A$ is a learnable weight matrix, and $\hat{A}_{ij}$ represents the weighted cosine similarity between nodes $i$ and $j$. Given that there are few unidirectional effects between patients, except in epidemic scenarios, the learned adjacency matrix $\hat{A}$ is symmetric, aligning with realistic expectations of patient population graphs.

Typically, a realistic adjacency matrix is non-negative and sparse. However, $\hat{A}$ represents a fully connected graph, which is computationally expensive, with elements ranging from $[-1, 1]$. To address this, a non-negative sparse graph $A$ is derived from $\hat{A}$ by applying the ReLU function, setting negative values to zero. This results in a latent graph $A$ suitable for downstream tasks.

*Graph Regularization.* Graph learning significantly influences the performance of graph SSMs in downstream tasks due to their sensitivity to graph structure. The sparsity, connectivity, and smoothness constraints of the learned graph are crucial for adaptive graph learning. The Dirichlet energy measures the smoothness of a set of graph signals $\{h_1, h_2, \cdots, h_N\}$:

$$\mathcal{L}_{\text{smooth}} = \frac{1}{2N^2} \sum_{i,j=1}^{N} A_{ij} \|h_i - h_j\|^2, \quad (3)$$

Equation (3) shows that as the distance between $h_i$ and $h_j$ decreases, the value of $A_{ij}$ increases. Consequently, the smoothness loss $\mathcal{L}_{\text{smooth}}$ promotes connections between similar nodes, enhancing the smoothness of graph signals on the learned graph $A$. Additionally, $\mathcal{L}_{\text{smooth}}$ helps regulate the sparsity of $A$. However, relying solely on $\mathcal{L}_{\text{smooth}}$ can lead to a trivial solution (i.e., $A = 0$). To

prevent this, two additional regularization terms are applied to $A$:

$$\mathcal{L}_{\text{con}} = \frac{-1}{n}\mathbf{1}^{\top} \log(A\mathbf{1}) \quad \text{and} \quad \mathcal{L}_r = \frac{1}{n^2}\|A\|_F^2, \qquad (4)$$

where $\mathcal{L}_{\text{con}}$ employs a logarithmic barrier to maintain the connectivity of $A$, while $\mathcal{L}_r$ prevents excessive sparsity induced by $\mathcal{L}_{\text{smooth}}$. The total graph regularization loss is defined as $\mathcal{L}_g = \mathcal{L}_{\text{smooth}} + \alpha \mathcal{L}_{\text{con}} + \beta \mathcal{L}_r$. Given the modality-aware representations $\mathbf{H}$, a graph structure $\mathbf{A}$ that meets the criteria of sparsity, connectivity, and smoothness can be obtained by minimizing $\mathcal{L}_g$:

$$\mathbf{A}^* = \arg\min_{\mathbf{A}} \mathcal{L}_g(\mathbf{A}, \mathbf{H}), \qquad (5)$$

## 2.3 Graph State Space Models

In this section, let $G = (V, E)$ represent a learned patient graph in Section 2.2, where $V = \{v_1, \ldots, v_n\}$ is the set of patient nodes and $E \subseteq V \times V$ is the set of similarity edges. Each node $v \in V$ is associated with a feature vector $\mathbf{h}_v^{(0)} \in \mathbf{H}$ in Section 2.1. For any $v \in V$, define $\mathcal{N}(v) = \{u \mid (v, u) \in E\}$ as the set of neighbors of $v$. For a subset of nodes $S \subseteq V$, $G[S]$ denotes the induced subgraph formed by the nodes in $S$, and $\mathbf{H}_S$ represents the feature matrix of the nodes in $S$.

*Tokenization.* Tokenization, the process of converting a graph into a sequence of tokens, is essential for adapting sequential encoders to graph structures. This section introduces a simple yet effective neighborhood sampling method for each node, highlighting its benefits over existing subgraph tokenization techniques [1]. The core concept involves sampling subgraphs for each node that accurately represent the node's neighborhood structure and its local and global positions within the graph, followed by encoding these subgraphs to generate node representations [15].

Given a node $v \in V$ and two integers $m, M \geq 0$, for each $0 \leq \hat{m} \leq m$, $M$ random walks of length $\hat{m}$ are initiated from $v$. Let $T_{\hat{m},i}(v)$ for $i = 0, \ldots, M$ denote the set of nodes visited in the $i$-th walk. The token corresponding to all walks of length $\hat{m}$ is defined as:

$$G\left[T_{\hat{m}}(v)\right] = G\left[\bigcup_{i=0}^{M} T_{\hat{m},i}(v)\right], \qquad (6)$$

which represents the union of all walks of length $\hat{m}$. This can be interpreted as the induced subgraph of a sample of the $\hat{m}$-hop neighborhood of node $v$. Ultimately, for each node $v \in V$, the sequence $G\left[T_0(v)\right], \ldots, G\left[T_m(v)\right]$ serves as its corresponding tokens.

*Local Encoding.* Given a node $v \in V$ and its sequence of tokens (subgraphs), the subgraph is encoded using encoder $\phi(.)$. This results in $\mathbf{x}_v^1, \mathbf{x}_v^2, \ldots, \mathbf{x}_v^{ms-1}, \mathbf{x}_v^{ms} \in \mathbb{R}^d$:

$$\mathbf{x}_v^{((i-1)s+j)} = \phi(G[T_i^j(v)], \mathbf{X}_{T_i^j(v)}), \qquad (7)$$

where $1 \leq i \leq m$ and $1 \leq j \leq s$. In practice, the encoder can be an MPNN, such as GCN [11], GAT [26], or GIN [30], which encodes nodes based on a sampled set of walks into feature vectors that include node features and local structural information.

*Global Encoding.* SSMs are recurrent models that require ordered input, whereas graph-structured data lacks an inherent order and thus necessitates permutation equivariant encoders. To address this, the Mamba architecture is modified by incorporating two recurrent

scan modules to process data in both forward and backward directions. Given two tokens $t_i$ and $t_j$, where $i > j$ and indices denote their initial order, in the forward scan, $t_i$ follows $t_j$, thereby acquiring information about $t_j$ (which can be integrated into the hidden states or filtered by the selection mechanism). Conversely, in the backward scan, $t_j$ follows $t_i$, thus obtaining information about $t_i$.

In the forward pass module, let $\Phi$ represent the input sequence, where $\Phi$ is a matrix with rows $\mathbf{x}_v^{sm}, \mathbf{x}_v^{sm-1}, \ldots, \mathbf{x}_v^1$. Additionally, let $A$ denote the relative positional encoding of tokens. The forward pass is then expressed as follows:

$$\Phi_{\text{input}} = \sigma\left(\text{Conv}\left(\mathbf{W}_{\text{input}} \text{ LayerNorm}(\Phi)\right)\right),$$

$$\mathbf{B} = \mathbf{W}_\mathbf{B}\Phi_{\text{input}}, \quad \mathbf{C} = \mathbf{W}_\mathbf{C}\Phi_{\text{input}}, \quad \Delta = \text{Softplus}\left(\mathbf{W}_\Delta \Phi_{\text{input}}\right),$$

$$\overline{\mathbf{A}} = \text{Discrete}_\mathbf{A}(A, \Delta),$$

$$\overline{\mathbf{B}} = \text{Discrete}_\mathbf{B}(\mathbf{B}, \Delta),$$

$$\boldsymbol{y} = \text{SSM}_{\overline{\mathbf{A}}, \overline{\mathbf{B}}, \mathbf{C}}\left(\Phi_{\text{input}}\right),$$

$$\boldsymbol{y}_{\text{forward}} = \mathbf{W}_{\text{forward},1}\left(\boldsymbol{y} \odot \sigma\left(\mathbf{W}_{\text{forward},2} \text{ LayerNorm}(\Phi)\right)\right),$$

$$(8)$$

where the parameters $\mathbf{W}, \mathbf{W}_\mathbf{B}, \mathbf{W}_\mathbf{C}, \mathbf{W}_\Delta, \mathbf{W}_{\text{forward, 1}}$, and $\mathbf{W}_{\text{forward, 2}}$ are learnable. The function $\sigma(.)$ denotes a nonlinear activation function, such as SiLU. LayerNorm(.) refers to layer normalization. The state space model is represented by SSM, and Discrete$(\cdot)$ denotes the discretization process. The same architecture is employed for the backward pass, albeit with different weights, using $\Phi_{\text{inverse}}$ as the input. This input is a matrix whose rows are $\mathbf{h}_v^1, \mathbf{h}_v^2, \ldots, \mathbf{h}_v^{sm}$. Let $\boldsymbol{y}_{\text{backward}}$ denote the output of this backward module, from which the final encodings are obtained by:

$$\boldsymbol{y}_{\text{output}} = \mathbf{W}_{\text{out}}\left(\boldsymbol{y}_{\text{forward}} + \boldsymbol{y}_{\text{backward}}\right). \qquad (9)$$

## 2.4 Patient Subphenotyping

*Clustering.* Patients can be clustered into groups based on the final representations. The proposed approach consists of two main components: (1) an architecture designed to encode the cluster assignments $\mathbf{C}$, and (2) an objective function used to optimize these assignments [24]. The cluster assignments $\mathbf{C}$ are obtained through the output of a softmax function, making the (soft) cluster assignment differentiable:

$$\mathbf{C} = \text{softmax}(\boldsymbol{y}_{\text{output}}) \qquad (10)$$

The proposed optimization objective combines insights from spectral modularity maximization with a regularization term to avoid trivial solutions:

$$\mathcal{L}_{\text{subtyp}}(\mathbf{C}; \mathbf{A}) = -\frac{1}{2m} \text{Tr}\left(\mathbf{C}^{\top}\mathbf{B}\mathbf{C}\right) + \frac{\sqrt{k}}{n}\left\|\sum_i \mathbf{C}_i^{\top}\right\|_F - 1, \qquad (11)$$

where $\|\cdot\|_F$ denotes the Frobenius norm. The computation of $\text{Tr}\left(\mathbf{C}^{\top}\mathbf{B}\mathbf{C}\right)$ is decomposed into a sum of sparse matrix-matrix multiplication and rank-one degree normalization, expressed as:

$$\text{Tr}\left(\mathbf{C}^{\top}\mathbf{A}\mathbf{C} - \mathbf{C}^{\top}\mathbf{d}^{\top}\mathbf{d}\mathbf{C}\right), \qquad (12)$$

This decomposition reduces the time complexity of computing $\mathcal{L}_{\text{DMoN}}$ from $O(n^2)$ to $O(d^2n)$ per update, enabling efficient clustering for sparse graphs.

*Model Optimization.* A joint loss function is employed to guide the simultaneous optimization of all modules:

$$\mathcal{L} = \mathcal{L}_{\text{subtyp}}(\mathbf{C}; \mathbf{A}) + \lambda \mathcal{L}_g(\mathbf{H}; \mathbf{A}) \tag{13}$$

where $\mathcal{L}_{\text{subtyp}}$ represents the task-aware loss, and $\mathcal{L}_g$ signifies the graph regularization loss. The hyperparameters $\lambda$, $\alpha$, and $\beta$ are used to balance these loss terms.

## 3 EXPERIMENTS

### 3.1 Experiment Settings

*Baselines.* Based on the different modalities of data usage, we select serveral baseline methods. The first method, widely used, is the widely used K-Means clustering method based on demographic data and aggregated time-series variables. The second method includes sequence modeling using LSTM, and 1D-CNN. The third method is the GNN model based on graph modeling, including GCN [11], GAT [26] and GIN [30]. Furthermore, the proposed MGSSM-SAKI model is compared with state-of-the-art (SOTA) graph transformers, such as NAGphormer [2] and GPS [20].

*Metrics.* To validate the stability of the identified subphenotypes, we calculated the Davies–Bouldin Index (DBI, a measure of the average similarity of each cluster with its most similar cluster), and the Calinski–Harabasz Index (CHI, the ratio between the within-cluster dispersion and the between-cluster dispersion).

### 3.2 Performance Comparison

This subsection evaluates MGSSM-SAKI on SAKI subphenotyping to verify MGSSM-SAKI's ability to improve graph learning quality and robustness. We performed 10-fold cross-validation on MIMIC-IV. We report the DBI, CHI and standard deviation in Table 1, where the best ones are in bold and the runner-ups are underlined. As demonstrated in Table 1, our MGSSM-SAKI model exhibits superior performance, evidenced by the highest intra-cluster similarity and inter-cluster distinction. The superior performance of our model can be attributed to three main factors: (1) The design utilizes long sequences of tokens to learn node encodings, followed by a selection mechanism that filters out irrelevant nodes. This approach allows MGSSM-SAKI to capture long-range dependencies without encountering scalability or over-squashing issues. (2) The selection mechanism in MGSSM-SAKI effectively filters the neighborhood around each node, ensuring that only relevant information is incorporated into the hidden states. (3) The random-walk-based neighborhood sampling provides diverse samples of neighborhoods, capturing the hierarchical nature of $k$-hop neighborhoods [27].

### 3.3 Ablation Study

We consider the following configurations: (1) **w/o diagnoses:** remove diagnosis data; (2) **w/o LGI:** using a precomputed patient graph structure; (3) **w/o GNNs:** remove the local encodings; (4) **w/o bi-SSMs:** remove the bidirectional SSMs and use a simple SSMs. As shown in Table 2, we can conclude that: (1) removing diagnosis data slightly degrades performance, underscoring the value of diagnoses in capturing patient conditions. (2) Using a precomputed patient graph structure instead of adaptive latent graph inference leads to reduced performance, highlighting the advantage of learning graph

**Table 1: Comparison results of baseline methods on MIMIC-IV: "average DBI/CHI ± standard deviation".**

| Method | DBI ↓ | CHI ↑ |
|---|---|---|
| K-Means | 2.35 ± 0.05 | 460.67 ± 15.32 |
| LSTM | 2.24 ± 0.01 | 464.89 ± 12.45 |
| Transformer | 2.21 ± 0.03 | 470.12 ± 11.78 |
| 1D-CNN | 2.23 ± 0.05 | 467.34 ± 13.60 |
| GCN [11] | 2.15 ± 0.04 | 475.56 ± 14.22 |
| GAT [26] | 2.13 ± 0.03 | 470.78 ± 10.89 |
| GIN [30] | 2.09 ± 0.06 | 475.89 ± 15.00 |
| NAGphormer [2] | 2.01 ± 0.03 | 485.23 ± 11.25 |
| GPS [20] | 2.03 ± 0.04 | 485.36 ± 12.75 |
| MGSSM-SAKI (ours) | **1.95 ± 0.02** | **492.34 ± 10.22** |

structures tailored to specific tasks. (3) The absence of local encodings via GNNs results in the most significant performance drop, indicating the critical role of capturing local structural information for accurate patient subphenotyping. (4) Replacing bidirectional SSMs with simple SSMs leads to a moderate decline, suggesting the bidirectional approach's effectiveness in capturing dependencies.

**Table 2: Ablation study results.**

| Variants | DBI | CHI |
|---|---|---|
| Vanilla | **1.95 ± 0.02** | **492.34 ± 10.22** |
| w/o diagnoses | 2.03 ± 0.01 | 490.11 ± 12.25 |
| w/o LGI | 2.03 ± 0.03 | 484.34 ± 9.26 |
| w/o GNNs | 2.04 ± 0.03 | 489.10 ± 10.14 |
| w/o bi-SSMs | 2.01 ± 0.04 | 490.87 ± 9.32 |

## 4 CONCLUSION

To sum up, this paper presents MGSSM-SAKI, a subphenotype identification model for SAKI patients based on bidirectional graph SSMs. We introduce multimodal inputs and design an adaptive latent graph inference module to uncover intrinsic relationships among patients, thereby constructing an optimal graph structure. Then we implement a differentiable strategy to group SAKI patients and identify three significant subphenotypes. The proposed framework is highly effective in identifying early ICU admission subphenotypes, improving personalized management.

## ACKNOWLEDGMENT

This study was supported by Zhejiang Provincial Natural Science Foundation of China [LZ22F020014], National Natural Science Foundation of China [823720951], Social Science Project of Chinese Ministry of Education [22YJA630036] and Beijing Natural Science Foundation [7212201].

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

# APPENDIX

# A RELATED WORKS

In this section, we will discuss the following two lines of related work including deep learning-based SAKI subphenotyping, message-passing neural networks, graph Transformers and SSMs.

## A.1 Sepsis with Acute Kidney Injury Subphenotyping

Sepsis with acute kidney injury (SAKI) is a heterogeneous clinical entity with significant variability in patient outcomes. Several studies have explored the use of deep learning to identify subphenotypes of SAKI, leveraging large datasets from electronic health records (EHRs) and other clinical data sources.

[19] utilized deep learning to analyze EHR data from the MIMIC III database. They identified three distinct subphenotypes of SAKI with differing clinical outcomes using a deep learning approach combined with K-means clustering. Their findings indicated that the subphenotype with the lowest severity of illness markers had the lowest dialysis requirement and mortality rate 28 days post-AKI. [16] introduced biologically informed neural networks (BINNs) for enhanced biomarker discovery and pathway analysis. Their

study differentiated between two subphenotypes of septic AKI and COVID-19 using plasma proteome data. They demonstrated that metabolic processes and immune system functions were critical for distinguishing between the subphenotypes, offering deeper biological insights into the mechanisms underlying these conditions.

Despite the advancements in applying deep learning techniques to SAKI subphenotyping, several limitations persist in the current research landscape. First, many studies utilizing deep learning for SAKI subphenotyping rely heavily on single-source data. These approaches often overlook the rich multimodal information available in EHRs, including clinical notes and diagnoses [32]. The lack of integration of these diverse data types can limit the comprehensiveness and accuracy of the resulting subphenotypes. Second, current deep learning models often treat patient data in isolation, failing to adequately account for the similarities and relationships between patients [21]. Finally, these models often struggle with sparse and high-dimensional data, which is a common characteristic of clinical datasets [10]. The high dimensionality of EHR data, coupled with its sparsity (many missing or infrequent data points), poses significant challenges for effective model training and accurate prediction.

## A.2    Message-Passing Neural Networks

Message-passing neural networks (MPNNs) are a class of graph neural networks (GNNs) that iteratively aggregate local neighborhood information to learn node and edge representations. MPNNs have emerged as the dominant paradigm in machine learning on graphs, attracting significant attention and leading to the development of various powerful architectures, such as Graph Attention Networks (GAT) [26], Graph Convolutional Networks (GCN) [11], and Graph Isomorphism Networks (GIN) [30]. However, simple MPNNs face several major limitations: (1) their expressivity is limited to the 1-WL isomorphism test, (2) they suffer from over-smoothing, and (3) they are prone to over-squashing [30].

## A.3    Graph Transformers

The rise of Transformer architectures has significantly impacted various fields, including natural language processing and computer vision. Their adaptations for graph data have become popular alternatives to MPNNs. By utilizing global attention, Graph Transformers (GTs) treat every pair of nodes as connected, thereby addressing issues such as oversquashing and over-smoothing inherent in MPNNs [2, 20]. However, GTs possess a weak inductive bias and require appropriate positional or structural encoding to effectively learn graph structures. Consequently, numerous studies have concentrated on developing robust positional and structural encodings.

## A.4    State Space Models

State Space Models (SSMs), a type of sequence model, are typically recognized as linear time-invariant systems that map an input sequence $x(t) \in \mathbb{R}^L$ to a response sequence $y(t) \in \mathbb{R}^L$ [17]. SSMs utilize a latent state $h(t) \in \mathbb{R}^{N \times L}$, an evolution parameter $\mathbf{A} \in \mathbb{R}^{N \times N}$, and projection parameters $\mathbf{B} \in \mathbb{R}^{N \times 1}$ and $\mathbf{C} \in \mathbb{R}^{1 \times N}$ as follows:

$$\begin{aligned} h'(t) &= \mathbf{A}h(t) + \mathbf{B}x(t), \\ y(t) &= \mathbf{C}h(t). \end{aligned} \tag{14}$$

Solving the above differential equations in deep learning contexts is challenging. Therefore, discrete space state models discretize the system using a parameter $\Delta$:

$$\begin{aligned} h_t &= \overline{\mathbf{A}}h_{t-1} + \overline{\mathbf{B}}x_t, \\ y_t &= \mathbf{C}h_t, \end{aligned} \tag{15}$$

where

$$\begin{aligned} \overline{\mathbf{A}} &= \exp(\Delta\mathbf{A}), \\ \overline{\mathbf{B}} &= (\Delta\mathbf{A})^{-1}(\exp(\Delta\mathbf{A} - I)) \cdot \Delta\mathbf{B}. \end{aligned} \tag{16}$$

Discrete-time SSMs have been shown to be equivalent to the following convolution:

$$\begin{aligned} \overline{\mathbf{K}} &= (\overline{\mathbf{C}\mathbf{B}}, \overline{\mathbf{C}\mathbf{A}\mathbf{B}}, \dots, \overline{\mathbf{C}\mathbf{A}}^{L-1}\overline{\mathbf{B}}), \\ y &= x * \overline{\mathbf{K}}, \end{aligned} \tag{17}$$

which allows for efficient computation. Structured state space models offer an efficient alternative to attention mechanisms, enhancing the efficiency and scalability of SSMs through reparameterization. SSMs have demonstrated promising performance in time series data, genomic sequences, healthcare, and computer vision. However, they lack a selection mechanism, resulting in missed context.

Recently, an efficient and powerful selective structured state space architecture called MAMBA has been introduced [5]. MAMBA incorporates recurrent scans along with a selection mechanism to control which parts of the sequence influence the hidden states. The selection mechanism of MAMBA is interpreted as utilizing data-dependent state transition mechanisms, making $\mathbf{B}$, $\mathbf{C}$, and $\Delta$ functions of the input $x_t$. MAMBA has shown outstanding performance in language modeling, outperforming Transformers of the same size and matching Transformers twice its size. This has spurred several studies to adapt its architecture for various data modalities and tasks.

# B    OPTIONAL MODULES OF MGSSM-AKI

## B.1    Positional Encodings (PEs) & Structural Encodings (SEs)

To enhance the MGSSM-AKI framework, an optional step involves integrating structural and positional encodings into the initial features of nodes and edges. Positional encoding (PE) supplies information about a node's position within the graph, ensuring that nodes in proximity have similar PEs. Structural encoding (SE), conversely, provides insights into the subgraph's structure. In line with previous works [20], either the eigenvectors of the graph Laplacian or random-walk structural encodings are concatenated with the node features when PE or SE are required:

$$\mathbf{h}_v^{(\text{new})} = \mathbf{h}_v \| p_v, \tag{18}$$

where $p_v$ represents the positional encoding for node $v$. For consistency, $\mathbf{h}_v$ is used throughout the paper in place of $\mathbf{h}_v^{(\text{new})}$.

## B.2    Token Ordering

Adapting sequence models such as SSMs to graph-structured data necessitates an ordering of the tokens [5]. For simplicity, consider the case where $s = 1$. When $m \geq 1$, the architecture implicitly provides an ordered sequence. Specifically, for a given

node $v \in V$, the $i$-th token is sampled from the $i$-hop neighborhood of $v$, which includes all $j$-hop neighborhoods where $j \geq i$. Thus, with a sufficiently large $M$ (number of sampled random walks), $T_j(v)$ contains sufficient information about $T_i(v)$, but not vice versa. Consequently, the initial order is reversed, resulting in the sequence $\mathbf{x}_v^m, \mathbf{x}_v^{m-1}, \ldots, \mathbf{x}_v^2, \mathbf{x}_v^1$. This approach allows inner subgraphs to encapsulate information about the global structure. When $s \geq 2$, the same procedure is applied, reversing the initial order to $\mathbf{x}_v^{sm}, \mathbf{x}_v^{sm-1}, \ldots, \mathbf{x}_v^2, \mathbf{x}_v^1$. To ensure robustness against permutation of subgraphs with the same walk length $\hat{m}$, they are randomly shuffled. When $m = 0$, a critical issue arises regarding the ordering of nodes during node tokenization. Tokens must be ordered based on either their need for information about other tokens or their importance to the task. For nodes, particularly when long-range dependencies are significant, the former criterion is imperative. This challenge is addressed by the bidirectional scan process of our architecture, necessitating the ordering of nodes by their importance. Several metrics can measure node importance in a graph, such as various centrality measures, degree, $k$-core, and Personalized PageRank or PageRank. In our experiments, nodes are sorted by their degree to maintain efficiency and simplicity.

### B.3 Stacking

In practice, multiple layers of the bidirectional Mamba are stacked to achieve optimal performance. Due to the ordering mechanism, the final output state corresponds to a walk of length $\hat{m} = 0$, representing the node itself. Consequently, this final state encapsulates the updated node encoding.

### C MORE IMPLEMENTING DETAILS

### C.1 Parameter Settings

The training of models utilizes the Adam optimizer, employing a minibatch of 64 patients along with a cosine annealing schedule for the learning rate. Multi-modal data are embedded into a 512-dimensional space, and datasets are partitioned randomly into 10 subsets. The analysis averages results from a 10-fold cross-validation process, with seven subsets designated for training, one for validation, and two for testing in each iteration. The validation subsets aid in identifying optimal parameter values during training iterations, and the mean squared error loss function is applied for model training.

### C.2 Experimental Environment

Algorithmic implementation is performed using the PyTorch framework, and experimental procedures are executed on an NVIDIA GeForce GTX 3090 with 24GB of memory. GNN are implemented using the PyTorch Geometric Library.

### C.3 Dataset Introduction

*Database.* The Medical Information Mart for Intensive Care IV (MIMIC-IV) database was utilized, comprising deidentified electronic health records of patients admitted to the Beth Israel Deaconess Medical Center in Boston, Massachusetts, spanning the years 2008 to 2019. MIMIC-IV provides an integrated, anonymized, and comprehensive clinical dataset.

*Definition.* The sepsis ICD-9 codes utilized for this research are 995.91 and 995.92. To determine which patient has AKI, the conditions used are an increase in serum creatinine by $\geq 0.3$ mg/dL and an increase in serum creatinine to $\geq 1.5$ times baseline, which is known or presumed to have occurred within the prior seven days.

*Inclusion and Exclusion Criteria.* Patients under 18 or over 89 years old, those admitted for 24 hours or less, those with ESKD (End Stage Kidney Disease) or missing vital signs are excluded from the study. Additionally, patients who undergo dialysis within 48 hours after AKI diagnosis or die within that timeframe are excluded to avoid skewing the results with terminal patients. Only the first admission with AKI per patient is considered, even if multiple admissions occur during the study period. Laboratory values and vital sign measurements from admission to 48 hours post-AKI diagnosis are employed to identify subphenotypes.

*Data Processing.* Only features present in at least 70% of patients are included. Non-temporal missing values are imputed using K-nearest neighbor imputation. Subsequently, MinMax scaling is applied to the resultant data to standardize the feature values.

### D ADDITONAL EXPERIMENTS

### D.1 Visualization Analysis

Once the patient representations are obtained, student t-distributed Stochastic Neighbor Embedding (t-SNE) is employed to project the patient data into a two-dimensional space, enabling visual inspection for the identification of potential clusters. The results of the Voronoi tessellation are presented in Figure 2.

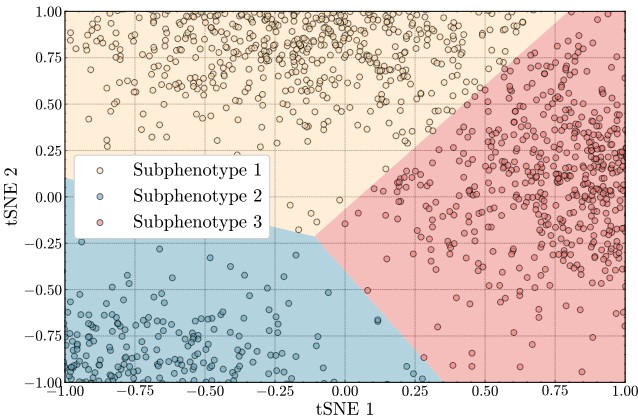

**Figure 2: The subphenotypes of SAKI are visualized using Voronoi tessellation. In this representation, each dot corresponds to an individual patient, forming clusters within a scaled-down, dimensionally reduced space generated by t-SNE technology. As a result, this space does not correspond to real-world units.**

### D.2 Interpretability Analysis

The distribution of variables across different subphenotypes is analyzed, and the results are presented in Table 3. Subphenotype 1 had

Table 3: Cluster descriptive statistics of SAKI subphenotypes.

| Clinical Feature | Subphenotype 1 ($n = 4,353$) | Subphenotype 2 ($n = 5,683$) | Subphenotype 3 ($n = 2,055$) | P Value |
|---|---|---|---|---|
| Male, $n$ (%) | 830 (58) | 1033 (54) | 424 (64) | <0.001 |
| Age (IQR) | 70 (58 to 79) | 66 (54 to 77) | 63 (52 to 73) | <0.001 |
| SAPS II Score, $n$ (%) | 38 (30–47) | 47 (37–57) | 54 (44–66) | <0.001 |
| Heart Rate (IQR) | 87 (75–103) | 91 (78–105) | 80 (74–83) | <0.001 |
| Respiratory Rate (IQR) | 20 (16–24) | 20 (16–25) | 21 (16–26) | <0.001 |
| Temperature (IQR) | 37.1 (36.5–37.7) | 36.9 (36.3–37.6) | 37 (36.4–37.6) | <0.001 |
| Systolic Blood Pressure (IQR) | 114 (101–131) | 109 (97–124) | 109 (96–124) | <0.001 |
| Diastolic Blood Pressure (IQR) | 58 (46–68) | 56 (48–67) | 56 (46–66) | <0.001 |
| Oxygen Saturation (IQR) | 97 (94–99) | 97 (95–99) | 97 (95–99) | <0.001 |
| White Blood Cell count (IQR) | 9.0 (8.1–13.6) | 6.6 (4.0–9.2) | 13.0 (7.3–14.2) | <0.001 |
| Blood Urea Nitrogen (IQR) | 22 (14–35) | 29 (17–48) | 32 (19–54) | <0.001 |
| Urine Output $n$ (%) | 1015 (70) | 749 (39) | 87 (13) | - |
| Serum Creatinine $n$ (%) | 102 (7) | 246 (13) | 76 (12) | - |
| Platelet count (IQR) | 232 (168–324) | 140 (63–236) | 126 (69–249) | <0.001 |
| Sodium (IQR) | 137 (134–140) | 136 (133–139) | 136 (133–139) | <0.001 |
| Potassium (IQR) | 4.0 (3.7–4.4) | 4.0 (3.7–4.5) | 4.2 (3.7–4.7) | <0.001 |
| Chloride (IQR) | 104 (100–108) | 106 (101–110) | 103 (99–108) | <0.001 |
| Bicarbonate (IQR) | 25 (23–28) | 22 (18–25) | 22 (18–26) | <0.001 |
| Phosphate (IQR) | 3.1 (2.5–3.9) | 3.4 (2.7–4.3) | 4.0 (3.1–5.3) | <0.001 |
| Hemoglobin (IQR) | 10.5 (9.4–11.9) | 9.7 (8.8–10.9) | 10.1 (9.0–11.3) | <0.001 |
| Lactate (IQR) | 1.6 (1.1–2.3) | 2.1 (1.4–3.3) | 3.6 (2.1–6.4) | <0.001 |
| Partial Thromboplastin Time (IQR) | 32.1 (27.5–42.42) | 35.2 (29.4–46.0) | 41.1 (32.8–57.6) | <0.001 |
| Anion Gap (IQR) | 13.0 (11.0–16.0) | 14.0 (12.0–17.0) | 16.0 (13.0–20.0) | <0.001 |
| Hematocrit (IQR) | 31.4 (28.2–35.3) | 28.7 (25.9–32.0) | 29.7 (26.8–33.2) | <0.001 |
| Partial Thromboplastin Time (IQR) | 14.6 (13.3–17.1) | 15.4 (13.8–18.8) | 16.7 (14.5–20.8) | <0.001 |

4,353 (36%) patients, subphenotype 2 had 5,683 (47%) patients, and subphenotype 3 had 2,055 (17%) patients. Patients in subphenotype 3 are the youngest, with a median age of 63 years (interquartile range [IQR], 52–73 years) compared to 66 years (IQR, 54–77 years) in subphenotype 2 and 70 years (IQR, 58–79 years) in subphenotype 1 (P <0.001). The Simplified Acute Physiology Score II (SAPS II) is highest in subphenotype 3 (54 versus 47 in subphenotype 2 and 38 in subphenotype 1; P <0.001). Minor but significant differences in blood pressure are observed among the three subphenotypes, with notable disparities in the proportions of patients requiring

vasopressor support (76% in subphenotype 3, 62% in subphenotype 2, and 39% in subphenotype 1; P <0.001). Unspecified septicemia is the primary discharge diagnosis across all three subphenotypes. Additionally, patients in subphenotype 3 exhibit worse kidney function parameters, including higher creatinine levels (1.6 versus 1.2 and 1.0 mg/dl; P <0.001), higher blood urea nitrogen (BUN) levels (32 versus 29 and 22 mg/dl; P <0.001), and lower bicarbonate levels (22 versus 22 and 25 mEq/L; P <0.001) compared to subphenotypes 2 and 1, respectively.