# OpenReview forum: "Identifying Subphenotypes for Sepsis with Acute Kidney Injury via Multimodal Graph State Space Models"
_KDD.org/2024/Workshop/AIDSH — KDD-AIDSH 2024 Poster_

### Official Review · Reviewer_5MFB · 2024-06-19

**Rating:** 6
**Confidence:** 4

**Review:**

This paper presents MGSSM-SAKI, a subphenotype identification model for SAKI patients based on bidirectional graphSSMs. The authors introduce multimodal inputs and design an adaptive latent graph inference module to uncover intrinsic relationships among patients, thereby constructing an optimal graph structure.Then the authors implement a differentiable strategy to group SAKI patients and identify three significant subphenotypes. The proposed framework is highly effective in identifying early ICU admission subphenotypes, improving personalized management.

Strenghs:
1. This paper presents MGSSM-SAKI, a novel Multimodal Graph Selective State Space Model for identifying subphenotypes of SAKI.
2. The efficacy of theproposed model is demonstrated through experiments conductedon the publicly available Medical Information Mart for IntensiveCare IV (MIMIC-IV) dataset.
3. The results indicate that it outperforms baseline models.

Weaknesses:
1. In the experiments section, RQ3 is not included in the main content of the paper but is identified as a primary question addressed by the experiments. Due to page limitations, I recommend either removing RQ3 from the experiments or relocating section D.2 to the main content.
2. The proposed method is too complex, but there is no computational complexity in the paper. I suggest the authors either show the running time or analyze the complexity of the proposed method.
3. The proposed framework is not novel. Each part of the proposed framework exists in existing literature. However, from the application aspect I agree with their contribution.

---

### Official Review · Reviewer_HaL5 · 2024-06-19
**Interesting method**

**Rating:** 6
**Confidence:** 4

**Review:**

Pros:

The pipeline is well-designed for healthcare-related problems, especially in handling data irregularities, fusing/tokenizing diverse data types, and modeling electronic health records (EHR) as a graph. This approach has potential to advance healthcare-related research by providing a robust framework for integrating and analyzing complex multimodal data.

Cons:

Ground truth labels and metrics: The reviewer challenges the absence of true labels for sub-phenotypes. Without true labels, it is unclear how the authors can accurately assess the performance of the proposed model. The reviewer disagrees with the use of DBI and CHI as proper metrics for measuring performance. These indices, while useful for clustering evaluation, may not fully capture the clinical relevance or accuracy of the identified sub-phenotypes. The authors should consider more clinically meaningful metrics or provide additional justification for the chosen evaluation methods.

---

### Decision · Program_Chairs · 2024-06-28

Accept (Poster)